# A Solver-Aided Hierarchical Language for LLM-Driven CAD Design

## Abstract

Large language models (LLMs) have been enormously successful in solving a wide variety of structured and unstructured generative tasks, but they struggle to generate procedural geometry in Computer Aided Design (CAD). These difficulties arise from an inability to do spatial reasoning and the necessity to guide a model through complex, long range planning to generate complex geometry. We enable generative CAD Design with LLMs through the introduction of a solver-aided, hierarchical domain specific language (DSL) called AIDL, which offloads the spatial reasoning requirements to a geometric constraint solver. Additionally, we show that in the few-shot regime, AIDL outperforms even a language with in-training data (OpenSCAD), both in terms of generating visual results closer to the prompt and creating objects that are easier to post-process and reason about.

## 1 Introduction

Parametric Computer-Aided Design (CAD) systems revolutionized manufacturing-oriented design by introducing a paradigm where geometry is created through a sequence of constructive operations. This approach enables both accuracy and precision in modeling and offers flexibility in design editing. Essentially, CAD systems use domain-specific languages (DSLs) to express geometry as a program, with CAD GUIs as end-user programming interfaces.

Recent advances in generative AI have significantly enhanced the creation of 2D and 3D geometry, yet achieving the precision, detail, and editability provided by CAD models remains a challenge. To bridge this gap, one promising strategy is to harness the powerful code generation capabilities of pre-trained large language models (LLMs) and the geometry-as-a-program paradigm from CAD. Rather than generating the geometries directly, we generate CAD programs that produce the geometric structures. However, this raises a crucial question: **How can we reimagine the traditional CAD DSL principles, which have been designed for a constant visual feedback loop, to craft innovative languages for design in an age where code is generated with support from AIs?**

In this work, we address this question and propose a new DSL for CAD modeling with LLMs, which we call AIDL: AI Design Language. Through experiments with different existing models and prior work that analyzes their observed behavior, we identify four key ***design goals*** for our DSL. Namely, we propose a *solver-aided approach* that enables LLMs to concentrate on high-level reasoning that they excel at while offloading finer computational tasks that demand precision to external solvers. For CAD, this means that the DSL should enable implicitly referencing previously constructed geometry (***dependencies***) and specifying relationships between parts that can then be solved by the solver (***constraints***). Further, we aim to create *semantically meaningful abstractions* that leverage the LLM's proficiency in understanding and manipulating natural language (***semantics***). Finally, we advocate for a *hierarchical design approach*, which allows for encapsulating reasoning within different model parts and enhancing editability (***hierarchy***).

Our analysis of existing CAD DSLs reveals that none achieve all four design goals, and supporting all goals simultaneously presents challenges due to conflicting requirements. For example, the ability to unambiguously reference all intermediate parts of the geometry (***dependencies***) is a known challenge in CAD. While recent work proposes a language that supports unambiguous referencing, it requires semantic complexity (***semantics***). Additionally, while constraints are widely used in specific aspects of CAD design, such as assembly modeling (***constraints***), supporting them in a complex model with hierarchically defined constraints (***hierarchy***) is computationally challenging.

Our key insight is that we can address these challenges by both limiting and expanding different language constructs from prior CAD DSLs. While we limit the use of references to *constructed* geometry, without losing geometric expressivity, we expand the use of constraints to hierarchical groups of geometry, so called *structures*. We support these novel language constructs with a recursive constraint solver that leverages the hierarchical structure to tractably solve global constraint systems.

We present a series of text-to-CAD results in 2D generated with our language, and we evaluate the importance of different aspects of AIDL by comparing it to OpenSCAD, a popular CAD language, and subsets of the AIDL language that has hierarchy or constraints disabled. For these methods, we report CLIP scores of the generated results and conducted a perceptual study on the generated CAD renderings. Our experiments show that AIDL programs are visually on-par with or better than their OpenSCAD counterparts despite the LLM not seeing AIDL code in its training data, while having superior editability, and our ablations demonstrate that introducing hierarchy contributes to local editability, while constraints allow complex multi-part objects to be composed precisely. With AIDL we show that language design alone can improve LLM performance in CAD generation.

```python
1  from AIDL import *
2
3  telephone = Solid()
4  base = Solid()
5  telephone.base = base
6  base.geometry = RoundedRectangle((0,0), base_length,
   base_width, 0.5*inch)
7
8  dial_plate = Solid(shift=(0, base_width * 0.15))
9  telephone.dial_plate = dial_plate
10 dial_plate.geometry = Circle((0,0), dial_radius)
11
12 receiver = Solid(shift=(0, base_width * 0.5))
13 telephone.receiver = receiver
14 receiver.left_end = Circle((-receiver_length / 2, 0),
   receiver_end_radius)
15 receiver.right_end = Circle((receiver_length / 2, 0),
   receiver_end_radius)
16
17 handle_length = receiver_length - 2 * receiver_end_radius
18 receiver.handle = RoundedRectangle((0,0), handle_length,
   handle_thickness, corner_radius=handle_thickness/2)
19
20 receiver.AddConstraint(Coincident(receiver.handle.left,
   receiver.left_end.center))
21 receiver.AddConstraint(Coincident(receiver.handle.right,
   receiver.right_end.center))
22
23 num_holes = 10
24 dial_plate.holes = [
25     Circle(
26         (dial_radius * 0.7 * cos(i * (2*pi / num_holes)),
27          dial_radius * 0.7 * sin(i * (2*pi / num_holes))),
28         0.2*inch
29     ) for i in range(num_holes)
30 ]
```

Figure 1: **A 2D CAD program in AIDL, generated using the prompt "old-school telephone".** The LLM generates AIDL code in a hierarchical fashion, adding constraints using naturally named operators. AIDL's backend solver produces the final CAD shape rendered on the right.

## 2 RELATED WORK

### 2.1 CAD GENERATION

The compilation of large CAD datasets in recent years (Koch et al., 2019; Willis et al., 2021b; Jones et al., 2021; Willis et al., 2022) has inspired a wealth of research on synthesizing CAD models. These efforts fall into two broad categories; those which generate CAD geometry directly (Willis et al., 2021a; Guo et al., 2022; Jayaraman et al., 2023; Nash et al., 2020; Xu et al., 2024; Liu et al., 2024), and those which generate a *procedure* that generates CAD geometry (Wu et al., 2021b; Ellis et al., 2017; 2018; Ganin et al., 2021; Ren et al., 2022; Li et al., 2023a; Xu et al., 2022; Lambourne et al., 2022; Para et al., 2021a; Seff et al., 2022; Willis et al., 2021b; Ma et al., 2024; Li et al., 2024; Khan et al., 2024). A fundamental challenge with these tools is the ability to control the generation. While many methods can be conditioned on an input allowing for reverse engineering

applications (Lambourne et al., 2022; Guo et al., 2022), the few methods that directly focus on generation give limited control over their output (Jayaraman et al., 2023; Wu et al., 2021a; Xu et al., 2024; Seff et al., 2022). The highest degree of control is afforded by those that take sketches as input, such as Free2CAD (Li et al., 2022) but these are effectively reverse reverse engineering an existing geometric design rather than enabling high level guidance. The goal of AIDL is to enable control without direct geometric supervision, and to incorporate semantic understanding beyond that of existing CAD programs. We have thus chosen to design our system around *general purpose* language models rather than CAD specific models, and focus on DSL design rather than the design or training of a generative model. Importantly, all prior works use CAD DSLs that have limitations when it comes to LLM needs, as we discuss in Section 3.1.

## 2.2 CODE GENERATION WITH LLMS

Software engineering has been one of the marquee applications of LLMs, so a detailed enumeration of works in the field is beyond the scope of this paper. We instead refer the reader to a survey Zhang et al. (2024), and reserve this section to position AIDL within the space. The majority of research on using LLMs for coding focus on how to make LLMs work more effectively with existing programming languages. A popular approach is to specifically train or fine-tune a model on code repositories and coding specific tasks (Li et al., 2023b; Lozhkov et al., 2024; Grattafiori et al., 2023), or more recently to use LLMs to generate higher complexity training examples (Xu et al., 2023; Luo et al., 2023). Other approaches tackle prompt complexity through system design, exploring prompt engineering and multi-agent strategies for pre-planning or coordinating a divide-and-conquer strategy (Dong et al., 2023; Bairi et al., 2023; Silver et al., 2023). AIDL approaches LLM code generation from an entirely different perspective, by asking which *language features* will best enable an LLM to work with a programming system. Most similar is BOSQUE, a proposed general purpose programming language (Marron, 2023). In particular, BOSQUE's embrace of pre and post conditions mirrors AIDL's use of constraints and strong validation, but does not go so far as to employ a solver to enforce constraints.

## 2.3 CAD DSLS

While there are many CAD DSLs, they can be grouped intro three broad categories:

**Constructive Solid Geometry (CSG)** In CSG, users can specify 2D and 3D parametric primitives, such as rectangles or spheres, directly in global coordinates. Using boolean operations, such as union or intersection, users then combine these primitives in a hierarchical tree structure to achieve complex designs. While some CSG languages, such as OpenSCAD, allow the use of variables or expressions for primitive parameters, they do not support specifying relationships or dependencies between different parts of the geometry. This absence of dependencies simplifies the abstraction, making CSG widely used in inverse design and reconstruction tasks (Du et al., 2018; Nandi et al., 2020; Yu et al., 2022; Michel & Boubekeur, 2021). However, this limitation also makes modeling more challenging, which is why CSG is not commonly used in most commercial CAD tools.

**Query-based CAD** Most commercial CAD tools use query-based languages, such as FeatureScript (Onshape, 2024), which employ a sequence of operators to create and modify models (e.g., extrude, fillet, chamfer). These operators reference intermediate geometry—e.g., a chamfer operator takes a reference to an edge. This referencing creates implicit dependencies, simplifying modeling and enabling easy editing as operations propagate when intermediate geometry is updated. However, a challenge arises when edits lead to topological changes, making reference resolution ambiguous. For example, if an edge gets split or disappears, where should the chamfer be applied? To address this, these languages do not reference geometry explicitly. Instead, geometric references are specified *implicitly* via a language construct called *queries*. These queries are resolved during runtime by a solver (CadQuery, 2024; Onshape, 2024), which typically uses heuristics to resolve ambiguities. This makes automating design challenging, and generative tools that use CAD operators restrict themselves to sequences where references are not needed, such as sketch and extrude (Wu et al., 2021a; Willis et al., 2021b; Lambourne et al., 2022). While recent work allows for the unambiguous direct specification of references (Cascaval et al., 2023), mastering this language is complex and demands significant expertise.

**Constraint-based CAD**   As the name implies, constraint-based CAD DSLs natively enable users to create geometric constraints between geometric primitives. This frees designers from specifying parameters consistently, allowing for freeform design while ensuring that relationships between parts are preserved. This approach is used in content creation languages like Shape-Assembly (Jones et al., 2020), GeoCode (Pearl et al., 2022), and SketchGen (Para et al., 2021b). In typical commercial CAD tools, constraint-based abstractions are used in sketches—2D drawings that get extruded to form 3D geometry—and during assembly modeling, but not during solid modeling which uses queries. These languages do not provide operations to modify primitives or to create intermediate geometry and therefore they reference geometry directly. Designs specified in these languages are non-hierarchical, all constraints are being solved simultaneously.

## 3   AIDL - A LANGUAGE FOR AI DESIGN

In this section, we present AIDL, a CAD DSL for LLM-generated designs.

### 3.1   LLM ANALYSIS AND DESIGN GOALS

We review the strengths and weaknesses of LLMs and formulate design goals that our DSL should support.

**Direct vs. indirect computation**   Findings by Bubeck et al. (2023) and Makatura et al. (2023) suggest LLMs perform better with external solvers. For CAD, we aim to enable LLMs to express design intent by specifying geometric relationships instead of performing direct computation. In modern CAD tools, geometric relationships can be defined using implicit dependencies or explicit constraints, each with trade-offs. Geometric dependencies create implicit constraints that are easy to enforce, but long chains of references are challenging to reason over (Makatura et al., 2023). Users typically avoid this issue by generating references automatically through CAD state interaction rather than writing CAD code directly. Explicit constraints, like those in CAD sketches or assemblies are easier to reason about, but harder to solve. It is also challenging to add just the right number of constraints so that the system is neither often under-or over-constrained. To achieve the best of both worlds, we aim to support both *implicit constraints through geometric dependencies (**dependencies**)* and *specification of geometric relationships via constraints (**constraints**)*.

**Named variables and semantic cues**   LLMs are designed to manipulate words, i.e., terms with semantic meaning. In their experiments, Makatura et al. (2023) reparametrize CSG programs with and without informing the LLM about the modeled object. Their results suggest that better reparametrizations are obtained by providing additional semantic knowledge. Our CAD DSL should use *intuitively named terms (**semantics**)* for design operations, references and constraints. Our language should also expose geometric entities easily, without many semantic indirections.

**Design complexity and modularity**   Bubeck et al. (2023) observe that GPT-4 can generate "syntactically invalid or semantically incorrect code, especially for longer or more complex programs." Similarly, Makatura et al. (2023) note that complex designs may miss components or have them incorrectly placed. To address this, our CAD DSL should treat *hierarchical design that supports modularity (**hierarchy**)* as a first-class construct, enabling the breakdown of complex problems into manageable units. This hierarchy should facilitate planning and iteration in code generation.

Table 1: We review how well the three major CAD DSL groups align with our design goals. Neither of the existing paradigms complies with all of the desiderata.

| Language | *dependencies* | *constraints* | *semantics* | *hierarchy* |
|---|---|---|---|---|
| CSG | - | - | ✓ | ✓ |
| Constraint-based | - | ✓ | ✓ | - |
| Query-based | ✓ | - | - | - |
| AIDL (Ours) | ✓ | ✓ | ✓ | ✓ |

None of the existing CAD DSLs fully support all of these design goals, as shown in Table 1. CSG DSLs are inherently hierarchical and can have intuitively named operations, but they do not support constraints, either implicitly through references or explicitly. Query-based DSLs allow implicit constraints via dependencies, but since references must be solved for though queries, they cannot be named directly, reducing semantic clarity. This also impacts modularity, as queries create chains of dependencies between distant parts of the program. Constraint-based CAD DSLs use intuitively named constraints, such as "coincident" or "symmetric," but they do not generate dependencies and lack hierarchy, as constraint solving is performed globally over a flat design.

## 3.2 KEY CHALLENGES AND DSL DESIGN DECISIONS

Combining all of the goals above in a single CAD DSL requires addressing two key challenges.

The first challenge is creating dependencies on previously constructed geometry (*dependencies*) without increasing the semantic complexity of operators (*semantics*). As explained in Sec. 2.3, previously constructed geometry cannot be persistently named because parametric variations often lead to topological changes. DSLs that reference previously constructed geometry use queries—algorithms that retrieve the geometry at a given state. However, this solution prevents assigning persistent semantic names to geometric entities, increasing semantic complexity and, our analysis shows that LLMs struggle to reason about queries with long chains, motivating our choice to disable them by design.

Our solution to enable dependencies without queries arises from the observation that all geometric primitives in CAD are created either through constructive operations that instantiate primitives or through boolean operations (e.g., when two edges intersect, a new vertex is generated). While this is evident for CSG DSLs we note that query-based CAD DSLs are not more expressive than CSG DSLs since all CAD operators (e.g. chamfering) can be expressed as a combination of a constructive and a boolean operation Cascaval et al. (2023). Reference challenges emerge from boolean operations, as changes in parameters can lead to a varying number of generated primitives.

While we still want the geometric expressivity enabled by boolean operations, we want to reference geometry without queries. To overcome this problem, we decide to restrict our DSL to only use references for geometry created before boolean operations. In our DSL, boolean operations are applied to *structures*, which is an intermediate type to create tree-structured hierarchies, see Fig. 5. The result of these booleans cannot be referenced, just as with CSG DSLs, however, we can reference *constructed* geometry and structures themselves. Although this introduces a language limitation, it does not affect 1) geometric expressivity, since in the worst case, you can have one geometry per structure, achieving the same expressiveness as CSG, and 2) dependency expressivity, as AIDL allows for arbitrary parametric expressions, meaning that in the worst case, dependencies can still be expressed manually, albeit with more effort.

Second, using constraints (*constraints*) to specify the relationship between elements within hierarchical designs (*hierarchy*) is computationally challenging. Hierarchical designs encourage growing complexity and an increasing number of constraints, driving down solver performance. Query-based languages deal with this complexity by solving constraints in intermediate, *flat* designs, e.g constraints between sketch elements in a CAD sketch are first solved before the user can extrude the sketch. Solving constraints from all CAD operations simultaneously is computationally too expensive for these systems. To tackle this challenge, we introduce (1) two types of constraints, one between geometry and one between *structures*, and (2) a custom recursive solver to hierarchically solve constraints in a design. This strategy allows us to explicitly define the hierarchy of constraints and to practically solve it, without providing intermediate feedback to the LLM.

## 3.3 AIDL BY EXAMPLE

Next, we showcase AIDL by example and show how the different language constructs fulfill our design goals. First, we will illustrate the basic constructs of AIDL with the phone handset example in Fig. 2. An AIDL program starts by defining the high-level logic of a design. These high-level building blocks are called structures and they are of different types, such as `Solid` and `Hole`, and they can be empty, a list of primitives, a list of substructures or any combination of these, see Fig. 5.

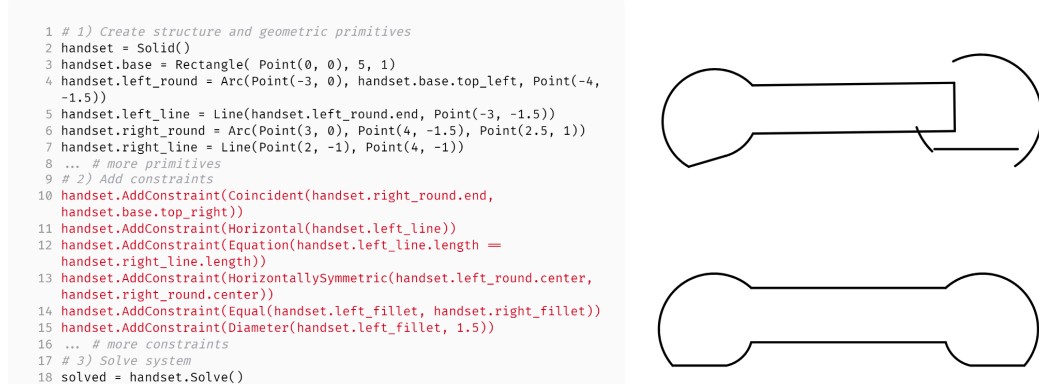

```
1  # 1) Create structure and geometric primitives
2  handset = Solid()
3  handset.base = Rectangle( Point(0, 0), 5, 1)
4  handset.left_round = Arc(Point(-3, 0), handset.base.top_left, Point(-4,
     -1.5))
5  handset.left_line = Line(handset.left_round.end, Point(-3, -1.5))
6  handset.right_round = Arc(Point(3, 0), Point(4, -1.5), Point(2.5, 1))
7  handset.right_line = Line(Point(2, -1), Point(4, -1))
8  ... # more primitives
9  # 2) Add constraints
10 handset.AddConstraint(Coincident(handset.right_round.end,
     handset.base.top_right))
11 handset.AddConstraint(Horizontal(handset.left_line))
12 handset.AddConstraint(Equation(handset.left_line.length ==
     handset.right_line.length))
13 handset.AddConstraint(HorizontallySymmetric(handset.left_round.center,
     handset.right_round.center))
14 handset.AddConstraint(Equal(handset.left_fillet, handset.right_fillet))
15 handset.AddConstraint(Diameter(handset.left_fillet, 1.5))
16 ... # more constraints
17 # 3) Solve system
18 solved = handset.Solve()
```

Figure 2: AIDL allows LLMs to express constraints using semantically meaningful operators. This figure demonstrates how adding constraints (highlighted in red) in an AIDL program for a phone handset eliminates geometrical flaws in the generated 2D sketch. (**Left**) AIDL code for handset design. (**Top right**) Design before constraints applied. (**Bottom right**) Design after constraints applied.

In the handset example, we first define an empty structure, L.2, which we populate with primitives, such as rectangles, lines and arcs, L.3-L.8. Next, we add unary and binary geometric constraints, e.g. Horizontal and Coincident, between these primitives, L.10-L.16. Finally, we solve the constraint system to optimize for the final parameters of each geometric primitive, L.18.

**References**    In AIDL, references are pointers to geometry, parameters or structures. They have various usages.

First, instead of specifying coordinates directly such as in L.3, we can use references to reuse already defined geometry. For example, in L.4, we define an Arc, which in the AIDL API is defined via `Arc(center, start, end)`. The `left_round` arc starts at the upper left corner of the `base` rectangle via the reference `handset.base.top_left`. This strategy lowers the risk of erroneously recomputing coordinates of the upper left point. Second, this reference ensures that `base` and `left_round` stay attached during the constraint solving process. Indeed, by sharing a common point, we *implicitly* define a coincidence constraint between them.

Geometric primitives can also be referenced within constraint calls. In L.10, we *explicitly* define a coincidence constraint between the upper right corner of `base` and the end point of the arc `right_round`. The arc `right_round` has been defined with explicit coordinates in L.6, which, without further constraints, is not necessarily connected to the rest of the shape, see Fig. 2 (top right).

Lastly, as can be seen in Fig. 5, references can also point to parameters of geometric primitives. This allows for more control and more expressivity when defining geometry and constraints. Consider L.12, where we used equation constraints to express a symmetric design intent on the two lines `left_line` and `right_line`. L.12 declares that both lines should have the same `length`, which is a parameter of the `Line` primitive. Parameters are referenceable on the same level as geometry and structures, making them first-class constructs in our language.

**Constraints**    Constraints express design intent, i.e., the way that geometry should behave under change. As we have already seen, in AIDL, constraints can be implied by sharing a reference, see L.4, or by explicitly adding them to the design via `AddConstraint` calls. Constraint operations have a certain constraint type and they take as input references. Depending on the constraint type, either equality or inequality constraints will be enforced on the geometric parameters specified by the input references. For example, in L.14, the `Equal` constraint type enforces the `diameter` of the two arcs `left_fillet` and `right_fillet` to be the same.

Using references and constraints, we can explicitly state the design intent, which will be realized by an external solver, L.18, (*dependencies*), (*constraints*).

**Synonymous operators**  References and constraints in a DSL are useful if they are easy to use. For human users, learning a new DSL can be challenging if its API is long and redundant. Concise APIs are usually preferred. However, designing a DSL for LLMs introduces a different criteria, which is that the LLM might write a function call which is not part of the API, but which is semantically equivalent. For example, consider the two constraint calls: (1) `AddConstraint(Perpendicular( line_1, line_2))` and (2) `AddConstraint(Orthogonal( line_1, line_2))`.

Intuitively, both `Perpendicular` and `Orthogonal` should enforce the same angle between the two lines, i.e., they are synonyms. However, to reduce redundancy, most APIs will choose only one of them. In AIDL, we expose both constraint types, to account for syntactical weaknesses of LLMs and to take advantage of their semantic versatility (***semantics***). More generally, we opt for a robust API vocabulary, allowing for different ways of constructing primitives, e.g. `Triangle(center, base, height)` vs. `Triangle(pt_a, pt_b, pt_c)`.

Note that even though we have synonymous references in AIDL, they are all being compiled to unique identifiers. During the interpretation of the program, we include only referenced entities in the model.

**Hierarchical designs**  Next, we illustrate the use of hierarchical designs with a complete phone design, see Fig. 1. The phone is an assembly made out of three different structures, the `base`, `receiver` and `dial_plate`, which are all `Solid` structures. These structures are directly attached to the `telephone` structure on lines 5, 9 and 13. As for the handset design in Fig. 2, each structure defines its own geometry and and constraints, e.g. the constraints for the receiver, L.20-21. Constraints can also be enforced between structures, which will be solved iteratively in tandem with structure-internal constraints, see Sec. 3.4.

Finally, in AIDL, the result of a boolean operation cannot be referenced, since the parameter-dependent topological outcome requires queries, see Sec. 3.1. To implement this, boolean operations are implied by using different structure types and then applied after constraint solving in a boolean post-process.

## 3.4   Compilation and Constraint Solving

The hierarchical organization of AIDL models allows for recursive constraint solving. We employ an iterative deepening, recursive solver strategy that allows AIDL to solve a minimal constraint problem at each stage, and also keeps substructures fixed as much as is possible to avoid unintuitive changes to substructures due to higher-level constraints. (translations of substructures are preferred over modification of internal geometry to satisfy constraints). To facilitate this recursive solving, AIDL models are first *validated* to ensure that each substructure is independently solvable, then *compiled* into a hierarchy of geometric constraint problems that we solve with an iterated Newton's method solver. The solved model is then *post-processed* to perform boolean operations and generate the final geometry.

When an AIDL program is run as a Python program, it generates a Structure tree data structure. An AIDL model is valid if Geometry only references other Geometry belonging to the same Structure, and Constraints only reference Geometry, Parameters and Structures within the same subtree. Definition of constraint equations in AIDL is *deferred* until after the tree structure is finalized because bounding boxes and some geometric constraints are not well defined until the model topology and initial parameters are fixed. Two non-inversion constraints are added to each bounding box, $height >= 0$ and $width >= 0$, using a slack variable formulation borrowed from linear programming (e.g. $height + s == 0 \land s - |s| == 0$).

The constraint system of an AIDL model is solved hierarchically as described in Appendix B using an iterated Newton's method solver (based on SolveSpace Westhues (2022)). Iteration is used to support bounding boxes; at each iteration we fix the expression of each bounding box limit relative to the initial positions of its geometry, then re-check and re-solve if a different piece of geometry now defines the limit. Solved AIDL models are post-processed to apply boolean operations defined by Solid and Hole Structures. Curve geometry is recursively aggregated to discover closed faces which are boolean unioned or subtracted from each other depending on the type of Structure they belong

to. We use the OpenCascade Modeling Kernel OCCT3D (2021) to perform boolean operations and generate output in the CAD standard STEP format.

## 4    EXPERIMENTS

**Implementation**    For our experiments, we perform LLM-driven 2D CAD generations with AIDL. AIDL enables LLM-driven text-to-CAD through a front-end generation pipeline. The pipeline follows a common **validate-until-correct** pattern. We first prompt the LLM with a detailed language description of AIDL, which includes AIDL's syntax, primitive geometry types, and available constraints. Then the LLM is prompted with six manually designed example programs in AIDL for these objects: bottle opener, ruler, hanger, key, toy sword, and wrench. Please refer to the supplemental material for the full list of prompts. Finally, it is prompted to generated the full AIDL program of the desired model. The front-end then executes the generated program, returning tracebacks directly to the LLM in case of failure and prompting the LLM to fix the error. This generation loop is repeated until either a syntactically correct program is found or after $N = 5$ failed attempts, taking advantage of incomplete executability to give feedback on partial generations. For all our experiments, we use the OpenAI's gpt-4o model without finetuning, and we run each prompt ten times with different seeds and collect the runs that generated a valid program.

**Results**    We report both the rendering and program of all runs of on 36 manually generated prompts in the supplemental material. In Figure 3, we show renderings for a diverse subset of the generated AIDL programs. Despite the LLM not being finetuned with our AIDL language, it successfully generates accurate CAD geometry based on its prior knowledge of these objects. Furthermore, the geometries are grouped hierarchically by semantically meaningful structures and constraints, making them easy to edit. See appendix D for an illustration of how an AIDL model can be modified.

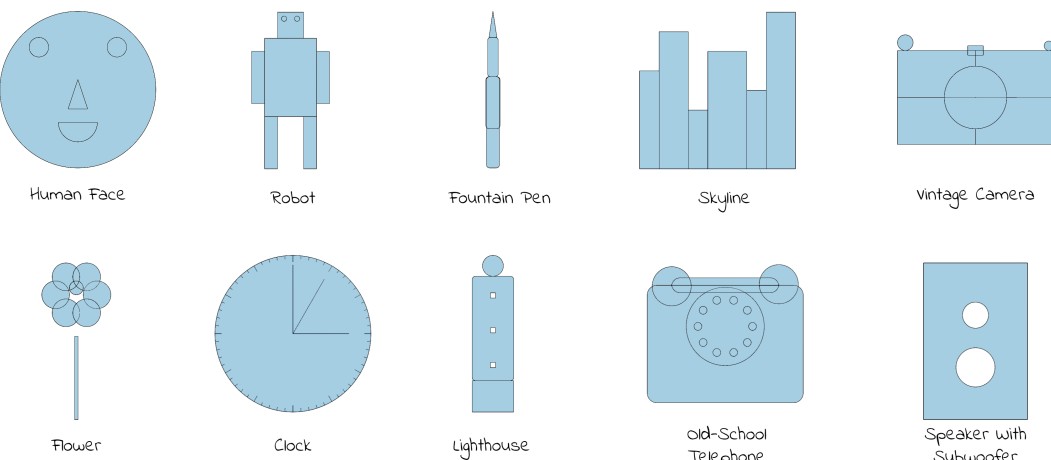

Figure 3: **A sample of LLM-guided 2D CAD generations using AIDL.** An untuned general purpose LLM is able to generate a diverse range of objects with accuracy after being prompted by the AIDL language syntax and a few example programs.

**Comparisons**    For comparison, we perform 2D text-to-CAD with the OpenSCAD language, the most common language for directly coding geometries in CAD, unlike other languages that are typically used with GUIs for end-user programming. We directly prompt the LLM to generated CAD geometry in the OpenSCAD language since the gpt-4o model has prior knowledge about its syntax. We used the same 36 prompts and report all results in the supplemental material. Despite the LLM's familiarity with OpenSCAD, we observe that AIDL results are often closer to the prompt and achieve higher CLIP scores (see Table 2). In addition to better prompt alignment, AIDL results exhibit more semantic structure. In particular, the OpenSCAD language does not support specifying relationships or dependencies between components, thus the LLM would often opt to generate polygons of the desired shape by specifying explicitly the vertex coordinates (see Figure 4), making the resulting program highly difficult to edit.

We also attempted using FeatureScript and the DSL from the recent work Cascaval et al. (2023) for LLM-drive 2D CAD generations. However, the LLM failed to generate syntactically correct programs in almost all cases. This issue was not rectified even when prompting the LLM with example programs and code documentations in those languages. These two languages are not syntactically based on common programming languages usually found in LLM training sets. This indicates the importance of designing a semantically rich language that is easy for the LLM to use and manipulate.

**Ablations**   We ablate our language design choices by comparing AIDL against two variants: $\text{AIDL}_{\text{no hierarchy}}$ and $\text{AIDL}_{\text{no constraints}}$, which disable hierarchy and constraints respectively. In $\text{AIDL}_{\text{no hierarchy}}$, all the geometries of a program will live on the same level under a single Structure instance, and all constraints will also be attributed to this single Structure. On the other hand, $\text{AIDL}_{\text{no constraints}}$ is a subset of AIDL where we have simply removed the ability to specify any constraints. For these language subsets, we modify our initial prompts to the LLM to reflect the altered language features. We report all runs on the same 36 prompts in the supplemental material.

While $\text{AIDL}_{\text{no constraints}}$ occasionally places components correctly, editing such programs is difficult because scaling requires individual adjustments for each component, whereas constraints allow a single edit to affect all geometries. Additionally, it often produces detached components due to the lack of constraints (see Figure 4 and the "fountain pen" example in the supplemental material).

Programs generated with $\text{AIDL}_{\text{no hierarchy}}$, while being visually similar to the ones generated with AIDL, are harder to refine, since the user cannot choose a particular part of the CAD shape to make local edits, as shown in Figure 4.

We observe that neither variation of AIDL significantly impacts CLIP scores for the renderings (Table 2), because that CLIP scores do not take into account editability and they place more emphasis on local semantics than having precisely connected geometries.

**Results Across Multiple Runs**   All methods produced at least one valid output per prompt, with success rates as follows: ours: 64%, $\text{AIDL}_{\text{no constraints}}$: 94%, $\text{AIDL}_{\text{no hierarchy}}$: 77%, and OpenSCAD: 79%. Notably, our method's success rate is only slightly lower than OpenSCAD, which is included in the training data. To showcase the highest-quality output for each method side by side, considering that LLMs produce varying outputs across runs, we conducted a perceptual study to rank the valid CAD programs generated from the 10 runs per method and prompt. The study details are discussed in appendix C, and the results are provided in the supplemental material.

**Limitations**   Our experiments revealed limitations of our system, particularly around model complexity and underused language features. AIDL supports rectangle rotation, yet all rectangles used in generated examples are axis-aligned. Looking at the generated code and conversation history (see supplemental) shows that the LLM did frequently specify that rectangles were rotatable (a flag in the Rectangle constructor), but failed to rotate them. One shortcoming of the AIDL library is that rectangles can only be rotated by the constraint solver, so an appropriate constraint (usually `Angle`) must be imposed to cause a rectangle to rotate. In cases where the LLM attempted to do this, it hallucinated a non-existent constraint like `Rotate` instead. When errors are reported to the LLM, the most common response is to try removing constraints or structures until the error goes away. Since we apply a validate-until-correct pattern, this means that the removed design intent (e.g. rectangle rotation) is never returned to the model. These limitations stem from our choice to focus on DSL design rather than the complementary approaches of model training or tuning, or prompt engineering. Fine-tuning a model on AIDL code could reduce the incidence of language feature hallucination, and crafting a more interactive prompting and feedback system could allow an LLM to recover lost complexity and design intent in the face of errors.

## 5   CONCLUSION

AIDL is an experiment in a new way of building graphics systems for language models; what if, instead of tuning a model for a graphics system, we build a graphics system tailored for language models? By taking this approach, we are able to draw on the rich literature of programming languages, crafting a language that supports language-based dependency reasoning through semantically meaningful references, separation of concerns with a modular, hierarchical structure, and that

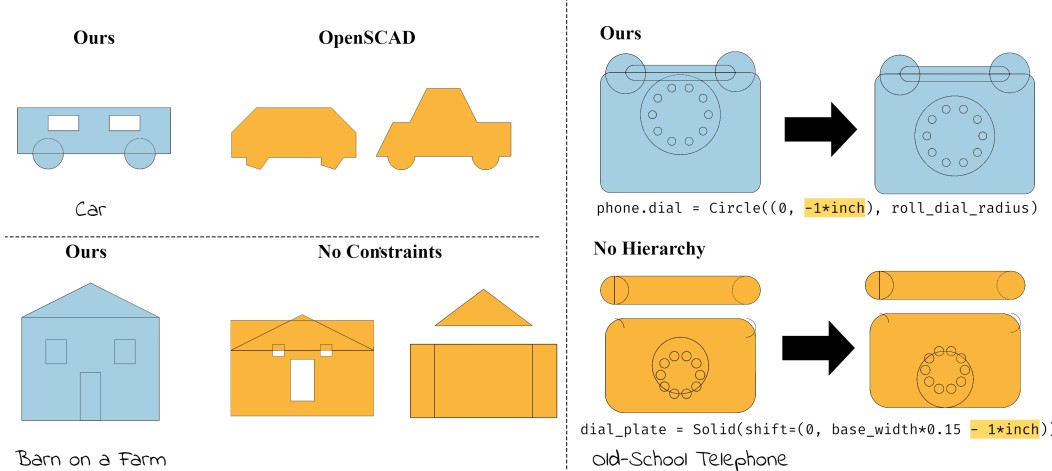

Figure 4: **Comparison and Ablation.** For the task of text-to-CAD, we compare our language to OpenSCAD and ablate on our language design choices. (**Top Left**) In particular, generated Open-SCAD programs exihibit manually drawn polygons with explicit vertex positions which are difficult to edit. (**Bottom Left**) Programs generated with **AIDL**no constraints has detached parts due to not being able to constrain the relative positions of part geometries. (**Right**) When an AIDL model is created with a structure hierarchy it is easier to locally edit because of modular substructures (left), while a similar edit on a non-hierarchical model (right) results in the model breaking (the dial moves without the dial holes). Performing the same edit in a non-hierarchical model requires multiple, non-concurrent edits.

Table 2: **Average CLIP scores for all prompts.** We perform text-to-CAD generation with **AIDL**, **AIDL**no hierarchy, **AIDL**no constraints, and **OpenSCAD** on our list of prompts for ten iterations each and show the average CLIP scores over the ones that produced valid programs.

|  | **AIDL** | **AIDL**$_{no\ hierarchy}$ | **AIDL**$_{no\ constraints}$ | **OpenSCAD** |
|---|---|---|---|---|
| ↑ **CLIP Score Avg.** | **28.90** | 28.64 | 28.89 | 27.32 |
| **CLIP Score Var.** | 2.24 | 1.98 | 2.05 | 1.87 |

compliments the shortcomings of LLMs with a solver assistance. Taking this neurosymbolic, procedural approach allows our system to tap into the general knowledge of LLMs as well as being more applicable to CAD by promoting precision, accuracy, and editability. Framing AI CAD generation as a language design problem is a complementary approach to model training and prompt engineering, and we are excited to see how advance in these fields will synergize with AIDL and its successors, especially as the capabilities of multi-modal vision-language models improve. AI-driven, procedural design coming to CAD, and AIDL provides a template for that future.

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

## A  LANGUAGE SYNTAX

| | | |
|---|---|---|
| *structure* | = | ⟨ *frame*, *sketch*, [*ref⟨structure⟩*], [*constraint*] ⟩ |
| *frame* | = | ⟨ *type* ∈ { Assembly, Solid, Hole, Drawing }, *orientation* ∈ { Top, Front, Side }, … ⟩ |
| *sketch* | = | ⟨ [*ref⟨geometry⟩*], [*ref⟨parameter⟩*] ⟩ |
| *parameter* | = | ⟨ val ∈ ℝ, mutable ∈ 𝔹 ⟩ |
| *ref⟨τ⟩* | = | ⟨ name ∈ String, ptr ∈ τ ⟩ |
| *geometry* | = | Point — Line — Arc — Circle — ⟨ [*ref⟨geometry⟩*], [*ref⟨parameter⟩*] ⟩ |
| *primitives* | ::= | make_point — make_line — make_arc — make_circle — make_rectangle — … |
| *constraint* | ::= | *logical_expr* — *structural_constraint* (*ref⟨τ⟩*, *ref⟨τ⟩*) |
| | — | *unary_geometric_constraint* (*ref⟨τ⟩*) — *binary_geometric_constraint* (*ref⟨τ⟩*, *ref⟨τ⟩*) |
| *structural_constraint* | ::= | above — center_inside — left_of — taller — … |
| *unary_geometric_constraint* | ::= | horizontal — diameter — fixed — ... |
| *binary_geometric_constraint* | ::= | coincident — tangent — equal — symmetric —... |
| *logical_expr* | ::= | *arith_expr* = *arith_expr* — *arith_expr* ≤ *arith_expr* — *arith_expr* ≥ *arith_expr* |
| | — | *logical_expr* ∧ *logical_expr* |
| *arith_expr* | ::= | c ∈ ℝ — *parameter* — *u_op arith_expr* — *arith_expr b_op arith_expr* |
| *u_op* | ::= | − — sin — cos — arcsin — arccos — sqrt — abs — norm — square |
| *b_op* | ::= | − — + — × — ÷ — min — max |

Figure 5: **Types and operations of AIDL.** $\tau$ represents the union type (structure—parameter—geometry). $[\theta]$ is the notation used to represent an array or list of $\theta$.

## B  SOLVER DETAILS

**Iterative Deepening Recursive Solve**   Constraint problems in AIDL are solved recursively over the structure tree in a post-order traversal, illustrated in the left half of Figure 6. At each step of this recursive solve, AIDL attempts to find a solution where only the geometry and parameters of the structure being solved, and *not* its substructures, are free parameters in the solve; everything deeper is initially treated as constants. This is done to minimize both the size of the constraint problem being solved, and to minimize perturbations to previously solved substructures. The validity condition that constraints can only reference geometry, structures, and parameters within a structure subtree ensures that if the constraints defined at the root of a subtree are satisfied, then the whole subtree is fully solved because child structure constraints cannot reference variables that would have changed.

Some constraint problems cannot be solved entirely locally, especially when a constraint in used to relate geometry between children. This is where we apply iterative deepening, in two stages. First we iteratively allow child structures at deeper levels to be translated by adding their translation frame parameters into the solver's set of free variables. As this search deepens, it also necessitates re-adding the constraint sets of the *parent* structures of translatable structures into the constraint set to be satisfied, since moving a child structure could invalidate a previously solved constraint. If translating structures is insufficient to satisfy the constraint system, then we repeat a similar iterative deepening, this time allowing all parameters, translation and otherwise to be solvable at each level. In this second iterative deepening it is necessary to include the constraints at the *same* level as the frontier of solver parameters, rather than the parent level, since geometric parameter changes could invalidate previously solved constraints. Iterative deepening continues until a valid solution is found, or all levels of the hierarchy have been exhausted (in which case the solve has failed because the constraint system is inconsistent or intractable.)

**Deferred Expressions**   While some constraints and expressions are well-defined mid-execution of an AIDL program, others can only be explicitly specified after the full topology and initialization of the model has been finalized by running the Python DSL code. The primary examples of these are bounding box coordinates, because they could depend on dynamically generated geometry, and ambiguous geometry constraints. An example ambiguous constraint is one like `Angle(L1, L2, theta)`, which constraints the angle between lines L1 and L2 to be equal to theta. The meaning of this constraint depends on the angle convention in use; is the angle measured clockwise or counter-clockwise between these two lines? In a traditional constraint language, a single consistent convention would be applied and programmers expected to learn and follow this convention, but a design principle of AIDL is to be flexible in calling conventions. To allow this, we *infer* the calling convention intended by picking the convention that is nearest to being satisfied by

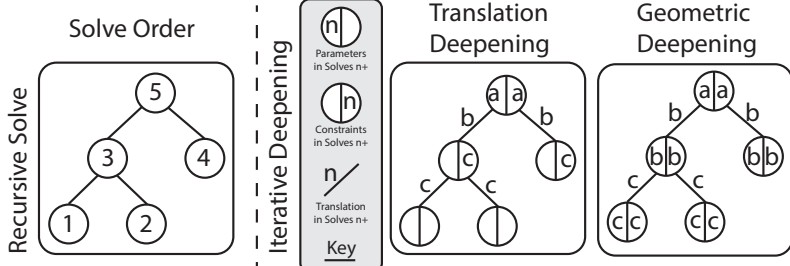

Figure 6: **Constraint solving order for an AIDL model. (Left)** The recursive solve order of the entire model. **(Right)** Iterative deepening of the constraint solver's scope for the root node (5 on left), in two stages, first translation deepening, then geometric deepening. Letters indicate the parameters and constraints included at each level attempted, and are accumulative within a stage (a, a and b, etc.)

the initial parameters of the constrained geometry. Since parameters are dynamically mutable, these determinations must also be deferred until immediately before constraint solving.

Bounding box expressions are also deferred until the context of their use in a constraint is known, and their exact formulation varies depending on which structures' bounding boxes are used in the same constraint. The rationale for this behavior is that constraints such as `struct.bb.top == struct.substruct.bb.top` leave an unbounded range for the sub-structure's top edge, since it is satisfied as long as that substructure has the highest top edge of any substructures. It is more likely that the intent of such a constraint is to align the top edge of a substructure with the top edge of its parent's sketch. To support this, bounding box expressions for structures coexisting in the same constraint expression as their descendants ignore those descendants' bounding boxes when computing the expressions for their coordinates.

**Iterated Newton Solve for Branching Expressions** AIDL expressions support the `min` and `max` operators, primarily to allow the use of bounding boxes. These create discontinuities in the constraint equation's Jacobians that use bounding box properties, which can cause a Newton solver to fail to converge. To combat this, we prune branches not used in constraint expressions given the pre-solve (initialization) parameter values, removing these discontinuities and increasing the chance of convergence. This effectively re-writes constraints to remove such functions: $\min(e_1, e_2) \rightarrow e_1$ (assuming $e_1 < e_2$ in the initial parameterization). The issue with this approach is that a solution to the re-written constraint problem may not be a solution to the original problem. We therefore check if the solution is valid for the original constraint problem and, if not, iteratively repeat this process using the rewritten constraint problem's solution as a new initialization until we find a valid solution.

## C  PERCEPTUAL STUDY

For our perceptual study, we presented users with all valid renderings of CAD programs generated for a particular prompt, asking them to select the best one for each method. Given the high number of prompts, the study was divided into four blocks, one for each method, with users randomly assigned to one block. We collected a total of 32 responses, with an average of 8 per method. The aggregated results are provided in supplemental material.

One limitation of this study was a small bug in the renderer that removed some lines from the images. While this compromised the results slightly, the study remains useful for observing differences across methods.

## D  EDITABILITY

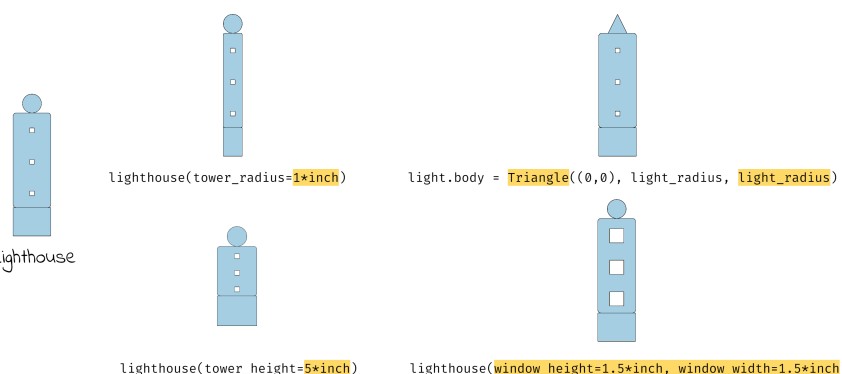

Figure 7: **Editability of AIDL.** Programs generated with AIDL have semantically meaningful parts. By changing the geometry of a single part in the original "lighthouse" **(left)**, we can modify the entire appearance of the CAD shape in various ways to produce a wide variety of semantically related, but visually distinct models.