# OpenReview forum: "A Solver-Aided Hierarchical Language For LLM-Driven CAD Design"
_ICLR.cc/2025/Conference — ICLR 2025 Conference Withdrawn Submission_

### Official Review · Reviewer_YStc · 2024-10-27

**Soundness:** 2
**Presentation:** 2
**Contribution:** 2
**Rating:** 5
**Confidence:** 3

**Summary:**

The paper presents AIDL (AI Design Language), a solver-aided hierarchical domain-specific language designed to enhance CAD modeling through the capabilities of large language models (LLMs).

Traditional CAD systems struggle with spatial reasoning and procedural geometry generation, which AIDL addresses by offloading complex spatial tasks to an external geometric constraint solver.

The authors identify four key design goals: enabling dependencies on previously constructed geometry, supporting explicit geometric constraints, leveraging the LLM's natural language understanding, and allowing hierarchical design for modularity.

Experiments demonstrate that AIDL outperforms existing CAD languages, such as OpenSCAD, in generating visually accurate and editable models, showcasing that thoughtful language design can significantly improve LLM performance in CAD applications.

**Strengths:**

- The methodology is well-structured and clearly articulated, allowing readers to easily follow the steps taken in the research.

- The central idea of the work is straightforward, making it accessible to a broad audience.

- The figures presented in the paper are highly effective in illustrating the main contributions of the research.

**Weaknesses:**

- The motivation for requiring a language description to identify the necessary objects is unclear. It is also questionable why a large language model (LLM) is needed to address this problem. For instance, why not leverage an LLM to search various websites for relevant raw CAD files based on specified keywords? Additionally, the discussion of the limitations of existing methods could be rewritten to more clearly articulate the specific challenges faced.

- The proposed method appears to be effective primarily for simpler examples compared to the existing capabilities demonstrated by OpenSCAD (see [OpenSCAD Demo](https://openscad.org/assets/img/screenshot.png)). The examples presented seem easily manageable through direct human editing "over the CAD object" or using the OpenSCAD software, raising concerns about the method's practical utility.

- Overall, the technological depth of this paper seems insufficient. Numerous studies have explored the reformulation of various tasks with the aid of LLMs. From my perspective, this paper presents yet another application of this idea without introducing significant advancements or insights.

**Questions:**

Could you help to address my concerns listed on the "Weakness" part?

---

### Official Review · Reviewer_MReW · 2024-10-31

**Soundness:** 1
**Presentation:** 2
**Contribution:** 2
**Rating:** 3
**Confidence:** 4

**Summary:**

The paper introduces AI Design Language (AIDL), a new hierarchical domain-specific language (DSL) for CAD design leveraging large language models (LLMs). It presents a novel approach for generating 2D CAD programs through hierarchical techniques, evaluated on 36 prompts with CLIP score as the evaluation metric.

**Strengths:**

1- Proposed a novel approach for generating CAD programs using hierarchical techniques.

2- Introduced a new application of LLMs for design tasks.

**Weaknesses:**

1- The paper evaluated the approach using only 36 prompts, making the dataset quite limited and insufficient for effectively evaluating LLMs.

2- Relying on the CLIP score may not provide an accurate evaluation for generated CAD designs. I strongly recommend creating a larger dataset with ground truth values that can support a more reliable evaluation.

3- The paper presents the results of the proposed approach but lacks a baseline or comparison with other methods in code generation.

4- There is no human evaluation conducted. Given the potential challenges in achieving precise automatic evaluation in this study, incorporating human evaluation would be valuable.

**Questions:**

1- Why does the paper generate 2D designs instead of 3D? The 2D designs resemble images rather than true CAD designs.

---

### Official Review · Reviewer_HdWK · 2024-11-03

**Soundness:** 3
**Presentation:** 3
**Contribution:** 2
**Rating:** 5
**Confidence:** 4

**Summary:**

The paper presents a promising approach to enhancing LLM-driven CAD design through the introduction of AIDL. The innovative integration of a geometric constraint solver and the focus on hierarchical, semantically rich language constructs are notable contributions. However, to strengthen the work, the authors should address the limitations related to performance analysis, error handling, and include user studies to validate the language's practical applicability.

**Strengths:**

1. AIDL effectively combines LLMs with a geometric constraint solver, enabling the generation of complex CAD models without requiring the LLM to handle intricate spatial reasoning. This approach allows for more accurate and semantically rich designs.
2. By incorporating hierarchical structures, AIDL facilitates modular design, making it easier to manage and edit complex models. This hierarchical approach aligns well with designers' workflows, improving the practicality of LLM-generated CAD models.
3. The experiments show that AIDL outperforms OpenSCAD in generating models that are closer to user prompts and are more editable. This is significant because OpenSCAD is included in LLM training data, whereas AIDL is not, highlighting the effectiveness of the language design.

**Weaknesses:**

1. The paper lacks a detailed analysis of the computational overhead introduced by integrating an external constraint solver. There are no benchmarks or discussions on how solver performance scales with model complexity, which is crucial for assessing practicality.
2. The approach relies heavily on the LLM's ability to generate correct AIDL code based on prompts. Without fine-tuning or extensive training data, there may be inconsistencies or errors in code generation, affecting the system's reliability.

**Questions:**

1. Has the computational efficiency of AIDL been benchmarked, especially concerning the constraint solver's performance with increasing model complexity?
2. Since LLMs can produce syntactic or semantic errors in code generation, what mechanisms does AIDL have to handle such errors, and how does it impact the overall system reliability? This is important for understanding the system's robustness.
3. Given that the experiments focus on a limited set of 2D models, how well does AIDL scale when generating more complex or detailed designs?

---

### Official Review · Reviewer_bSur · 2024-11-09

**Soundness:** 3
**Presentation:** 3
**Contribution:** 2
**Rating:** 3
**Confidence:** 5

**Summary:**

This paper proposes a hierarchical domain-specific language (DSL)  for modeling Computer-Aided Design Applications using LLMS. The idea is to use LLMs for high level reasoning while spatial and geometric reasoning is outsourced to a domain-specific solver. The evaluation compares different aspects of the proposed DSL with OpenSCAD.

**Strengths:**

1. The first DSL for CAD modeling using LLMs
2. In few-shot regime, AIDL outperforms OpenSCAD

**Weaknesses:**

1. How is it different from tool learning? In this case the tool is the solver. In fact you can consider multiple solvers.
2. Apart from providing a UI, it is not clear what reasoning is carried out by the LLM. It seems to me that the function of the LLM is to compile the constraints that will be solved by the solver. Can you elaborate on the reasoning tasks carried out by the LLM? The use of LLMs is essentially as a code generation tool in a particular domain. Where is the innovation?  Can you elaborate how it is different from code generation in a particular domain?
3. I didn't see any discussion on how to prevent errors being introduced by the LLM. CLIP scores  or the perceptual study will not provide any intuition about the  behavior of the LLM.  Better evaluation methods are needed as well as techniques to prevent bugs induced by the LLM (can an SMT solver be used?).

**Questions:**

1. I think the innovation in the paper has not been spelt out. In particular how is it different from code generation in a particular domain which is a well studied subject
2. Can something like an SMT solver be used verify the constraints (code) generated?
3. Are there better evaluation metrics? For example, the productivity of a designer using AIDL as opposed to a traditional CAD engine.

---

### Note · Authors · 2024-12-04

**Comment:**

Thank you to the area chairs and reviewers for your thoughtful consideration and feedback.

We appreciate that reviewers liked the novelty of the approach and articulation of our methodology, as well as the suggestions for improvement around stronger automatic and human mediated evaluation. We see that we must articulate more clearly the difference between this work and work in LLM tool use and code generation for domain-specific code. While most works in those fields focus on how to apply an LLM to a domain with a given language, this work explores how the design features of a language affect the ability of an LLM to work with it. By building languages to complement the strengths and weaknesses of a general purpose language model, we can avoid the time and cost associated with compiling large datasets and training or tuning a purpose-built model.

We are withdrawing our paper from consideration, and will use the insights from the review process as we improve our system, evaluation, and exposition for a future submission.

**Withdrawal Confirmation:**

I have read and agree with the venue's withdrawal policy on behalf of myself and my co-authors.